# Tailor-Made Immunochromatographic Test for the Detection of Multiple 17α-Methylated Anabolics in Dietary Supplements

**DOI:** 10.3390/foods10040741

**Published:** 2021-04-01

**Authors:** Barbora Holubová, Pavla Kubešová, Lukáš Huml, Miroslav Vlach, Oldřich Lapčík, Michal Jurášek, Ladislav Fukal

**Affiliations:** 1Department Biochemistry and Microbiology, University of Chemistry and Technology Prague, Technická 5, 166 28 Prague 6, Czech Republic; paja.kuba@centrum.cz (P.K.); miroslav-vlach@seznam.cz (M.V.); ladislav.fukal@vscht.cz (L.F.); 2Department of Chemistry of Natural Compounds, University of Chemistry and Technology Prague, Technická 5, 166 28 Prague 6, Czech Republic; lukas.huml@vscht.cz (L.H.); oldrich.lapcik@vscht.cz (O.L.); michal.jurasek@vscht.cz (M.J.)

**Keywords:** immunochromatography, stanazolol, anabolic steroids (AAS), 17α-methylated AAS, dietary supplements

## Abstract

In recent years, the undeclared presence of various anabolic androgenic steroids (AAS) in commercial supplements has been confirmed. This fact can be a potential threat to all athletes using these supplements, and therefore, there is of increased interest in the implementation of rapid methods for the detection of AAS. The presented study describes the development of an immunostrip test for the detection of multiple 17α-methylated AAS based on direct and indirect competitive principle using gold nanoparticles as a label. As a capture reagent on test lines conjugated stanazolol to rabbit serum albumin (RSA/ST-3) was used, the intensity of color formed in the test line of the AAS-positive sample was visually distinguishable from that of negative sample within 10 min. The optimized closed direct and indirect format of the test provided a similar visual detection limit (0.7 and 0.9 ng/mL, respectively). The most commonly orally abused AAS (17α-methyltestosterone, methandienone, methyldihydrotestosterone, oxandrolone and oxymetholone) showed a strong cross-reaction. Developed immunostrips were successfully applied to analysis of artificially contaminated dietary supplements with 17α-methylated AASs. The developed immunostrips offer potential as a useful user-friendly method for capturing suspicious dietary supplement samples with different contents of AAS at levels far below the usually used concentrations of AAS.

## 1. Introduction

In recent years, the use of anabolic androgenic steroids (AAS) has been increasing, despite their proven negative effects. AAS increase the physical endurance and performance of athletes, which might lead to better results in sports. Other users take AAS for purely aesthetic reasons for a more effective increase in muscle mass. Nevertheless, whether they are professional or amateur athletes, the undeclared presence of AAS [1,2] in dietary supplements pose a potential threat to all of them, whether for health or doping reasons [3,4,5].

The basic methods for the detection of AAS include, in particular, gas or liquid chromatography combined with mass detection (GC-MS, HPLC-MS, respectively). Other commonly used methods include enzyme-linked immunosorbent assay (ELISA) [6,7,8,9]. In our recent work by Huml et al. [7], we have described the development of a group-selective immunoassay method based on stanazolol (ST)-derived polyclonal rabbit antibodies (RAb). In the latter paper [7], hapten ST-3 obtained by alkylation of the pyrazole ring showed the best performance in convenient ELISA assays. Conjugated hapten to rabbit serum albumin (RSA) was simply marked RSA/ST-3 (Appendix A). As the result of this work [7], we found that the RAb 212 antibody had strong cross-reactivity (CR) for 17α-methylated AAS in an indirect competitive ELISA setup (Figure 1). The applicability of this group-selective method was further demonstrated in a series of police-intercepted preparations containing AAS.

Based on these results, our group began to develop an immunochromatographic test (ICT) that would be well applicable in field conditions and could be useful for the immediate identification of suspect samples. The ICT (or LFIA—lateral flow immunoassay) method is based on test strips, which consist of several parts (Figure 2A) [10]. ICT method was first used to determine human chorionic gonadotropin (hCG) in urine to detect pregnancy. Due to the ease of use and commercial attractiveness of immunochromatographic assays [11], research and development in this area has expanded considerably. A wide range of analytes including viruses (e.g., hepatitis B [12], HIV [13], SARS-CoV-2 [14]), bacteria (e.g., *Helicobacter pylori* [15], *Staphylococcus aureus* [16], *Salmonella* [17]), toxins [18,19,20], pesticides [21], cancer markers [22,23], antibiotics [24,25], drugs [26,27], terpenes [28] and steroids [29,30,31,32,33] have been used to construct useful immunochromatographic assays.

The assay is based on the interaction of antigen with antibody as they flow through the membrane. One of the binding partners is labeled with nanoparticles (NPs). The most commonly used colloid is gold (AuNPs), because it is the most stable, does not undergo decomposition processes by light and is non-toxic [34,35]. The method can be divided into open and closed formats (Figure 2A). Because the enclosed has the advantage that all immunoreagents are pre-immobilized on the test strips, and only a sample needs to be added, this method is very suitable for field use. The open format is mostly used for laboratory testing and method development. Immunoreagents are applied to the membrane to produce control (CL) and test (TL) lines. Based on the visibility of one or two bands after the reaction, the results can be read visually, or the bands can be evaluated by image analysis, e.g., scanned in degrees of color intensity and measured using a suitable computer program. The evaluation of the test is described in (Figure 2B).

ICT methods have a lower sensitivity compared to the ELISA, usually increasing the detection limit by one order of magnitude [21,36,37]. Therefore, in this work, the competitive test was arranged in indirect (Figure 2C) and direct format (Figure 2D), because the method of arrangement could have the effect of reducing the detection limit of the developed method. The difference is in the labeled antibodies. In the direct format, the primary antibody (against the analyte conjugate) is labeled, while in the indirect format, a tertiary antibody is bound to the NPs, which specifically binds to the primary antibody. 

In the current work, we describe the optimization of the composition of the strip test, its applicability on the detection of AAS, and successful testing of artificially contaminated dietary supplements with 17α-methylated AAS.

## 2. Materials and Apparatus

### 2.1. Immunoreagencies

Donkey anti-goat (DAG) antibodies, donkey antibodies against goat serum immunoglobulins, and goat anti-rabbit (GAR) antibodies, goat antibodies against rabbit serum immunoglobulins were purchased from Nordic Immunological Laboratories (Eindhoven, Netherlands); rabbit polyclonal antibodies against conjugate ST-3 (RAb 212) (primary antibody) and immobilization conjugate RSA/ST-3 were available from our previous work (Appendix A) [7].

### 2.2. Chemicals

Bovine serum albumin (BSA), polyethylene glycol 1500 (PEG) and Tween 20 were purchased from Merck (Kenilworth, NJ, USA). Trehalose was purchased from Glentham Life Science Ltd. (Wiltshire, UK). Steroid standards were purchased from Steraloids Inc. (Newport, RI, USA) and Sigma Aldrich Corp. (St. Louis, MO, USA). Steroids were dissolved in ethanol (96%) to a concentration of 1 mg/mL and stored at −18 °C.

### 2.3. Material for ICT

Enclosures for closed ICT format were purchased from Kinbio Tech Co., Ltd. (Shanghai, China). The membranes and pads used are summed-up in Appendix A. 

### 2.4. Buffers

Borate buffers; 0.2 mol/L, pH 8.8: 22.88 g of borax, 0.3 L of deionized water; 0.1 mol/L, pH 8.8: 0.5 L of 0.2 mol/L borate buffer, 0.5 L of deionized water, 5 mL 2% sodium azide; 5 mmol/L, pH 8.8: 5 mL of 0.1 mol/L borate buffer, 95 mL of deionized water, 1 mL of 2% sodium azide; carbonate–bicarbonate buffer, pH 9.6; 1.59 g of sodium carbonate, 2.93 g of sodium bicarbonate, 1 L deionized water; wash buffer for a solution of AuNPs with antibodies (5 mmol/L borate buffer, 1% BSA): 10 mL of 5 mmol/L of borate buffer, 0.1 g BSA; reaction buffer for ICT (0.1 mol/L borate buffer, 1% BSA, 1% PEG, 1% Tween): 10 mL of 0.1 mol/L borate buffer, 0.1 g BSA, 0.1 g, PEG, 100 μL 10% Tween 20; storage buffer for a solution of AuNPs with antibodies (0.1 mol/L borate buffer, 1% BSA): 10 mL of 0.1 mol/L borate buffer, 0.1 g BSA; drying buffer for ICT (0.2 mol/L borate buffer, 0.1% BSA, 3% trehalose, 1% Tween 20): 10 mL of 0.2 mol/L borate buffer, 0.01 g BSA, 0.3 g trehalose, 100 μL 100 Tween 20.

### 2.5. Apparatus and Software

CAMAG^®^ Linomat 5 (Muttenz, Switzerland) and programmable strip cutter ZQ2002 (Kinbio Tech Co. Ltd., Shanghai, China); TotalLab program (Newcastle-upon-Tyne, UK).

## 3. Experimental Section

### 3.1. Preparation of Conjugates of GAR-AuNPs and RAb-AuNPs

The individual steps for preparing the gold nanoparticle antibody conjugate (GAR-AuNPs) used in this work are clearly shown in Appendix A. GAR tertiary antibody diluted in redistilled water to a concentration of 1 mg/mL, AuNPs and 5 mmol/L borate buffer was used to prepare the conjugate (direct arrangement). The ratio of these reactants in the mixture was as follows: 2 mL of 5 mmol/L borate buffer; 1 mL of AuNPs; 100 μg of antibody. The same procedure for preparing a solution of AuNPs with antibody (RAb-AuNPs) for the direct arrangement was used (Appendix A). As to the indirect arrangement, the same ratio of reactants was used. The only difference was in the selected antibody; in this case, the primary antibody RAb 212 was used.

### 3.2. Preparation of Samples and Matrices

Matrices (food supplements without the presence of AAS; verified by gas chromatography according to Stepan et al. [38]) and samples of food supplements (obtained from the B&M Fitness Center Pelhřimov, 3D Fitness Academy Prague and the Fitness007 Prague store) were prepared as follows: 1 mL of EtOH was added to the 0.1 g matrix/sample weighing or 0.1 mL liquid sample, the mixture was shaken for 20 min using a multi-speed vortex and then stored at 4 °C. To test the effect of dietary supplements on both tests, diluted ethanolic matrix extracts were applied. For ICT, they were diluted 10×, 50×, 100× and 500× in reaction buffer. To contaminate the food supplements with other AAS samples, a protein matrix (Ultrafield 100% whey protein, Scitec Nutrition) was chosen. The matrix (399 mg) was contaminated by the addition of AAS as follows (see Figure 5): Mixture 1: nandrolone (0.33 mg), methandienone (0.33 mg) and ST (0.33 mg). Mixture 2: nandrolone (0.33 mg), oxymetholone (0.33 mg) and ST (0.33 mg). Mixture 3: nandrolone (0.33 mg), testosterone propionate (0.33 mg) and ST (0.33 mg). Mixture 4: testosterone enanthate (0.5 mg) and ST (0.5 mg). Mixture 5: testosterone propionate (0.5 mg) and ST (0.5 mg). Control: boldenone (1 mg), dihydrotestosterone- DHT (1 mg) and ST (1 mg).

### 3.3. Preparation of ICT Strips

The membrane was cut from the supply roll and glued to the pad for better handling. For the open format, plastic reinforcement was used, for the closed format, laminate. After the membrane was reinforced, immunoreagents were applied to the membrane using a Linomat 5 instrument to form control (CL) and test lines (TL). Antibodies were immobilized in CL. In the case of the indirect format, DAG was diluted in 5 mmol/L borate buffer to 100 µg/mL; in the case of the direct format, GAR was diluted in the same buffer to 150 µg/mL. CL was applied for a closed format of 17 mm and for an open format of 19 mm from the bottom edge of the membrane. RSA/ST-3 conjugate diluted in carbonate–bicarbonate buffer to a concentration of 300 µg/mL for the indirect and 200 µg/mL for the direct format was immobilized on TL (13 mm for closed and 15 mm for opened format from the edge). The resulting amount of individual immunoreagents was 2 µL per 1 cm of the membrane (i.e., 0.8 µL per one strip). The membrane, thus, treated was incubated overnight at 37 °C. The next day, other components were glued to the reinforcement to the membrane. In the case of the open format, the glass fibers as a support for the sample and for the closed format, conjugation and absorption pads were added. The conjugation and sample pad were placed in drying buffer for 1 min for the closed format on the first day and dried for 1 h at 37 °C. Its components are necessary for the correct course of the test and for the stabilization of the conjugate of AuNPs and antibodies, which are then applied to the conjugation pad using Linomat 5. For indirect format, 3 µL of GAR-AuNPs and 2 µL of RAb 212 antibody per strip, for direct format 3.5 μL of RAb-AuNPs conjugate were applied. The plate was incubated at 37 °C until the next day after conjugate application. In both cases, the membrane, thus, assembled was chopped with a 4 mm wide strip cutter.

### 3.4. Open Format

The strips of cut membrane were glued to a wooden base. The upper end of the membrane was covered with an absorbent pad for better sample uptake. A mixture of 40 µL of reaction buffer, 40 µL of sample and, in the case of direct arrangement, 1.5 µL of RAb-AuNPs conjugate was applied to the glass fibers, 3 µL of GAR-AuNPs and 2 µL of RAb 212 in the indirect arrangement.

### 3.5. Closed Format

The strips of cut membrane were fixed in plastic boxes. When fixing them, it was necessary to pay attention to the correct direction of the strips, and they should not bend. Subsequently, the box was closed. A mixture of 40 µL of sample and 40 µL of reaction buffer was applied to the lower window, then the test result could be observed in the second window (Figure 2A, closed format).

### 3.6. Test Evaluation

Both types of assays were evaluated visually by comparing the TL on the strip (where a steroid standard or sample was applied) against TL on the strip, where the applied solution did not contain analyte. If the TL intensity on the test strip was weaker than that of the negative control, it was a positive result. The results could also be evaluated semi-quantitatively by visual comparison of the sample strip with strips with different known analyte concentrations (calibration series) and, thus, to determine the approximate analyte concentration in the sample. Using the calibration series, the visual detection limit (VDL) can be determined, which is the lowest analyte concentration in the sample, reflected by the attenuation of TL on the strip to which the sample was applied against TL on the strip with no presence of the analyte. In some cases, the tests were also evaluated quantitatively using the computer program TotalLab, where the intensity of the lines was evaluated on the basis of grayscale. Their values were then plotted against the decimal logarithm of the target analyte concentration (ST). The resulting points were interpolated by a sigmoid curve (calibration curve), which can be characterized using a four-parameter regression equation (sigmoid):(1)$$S=C+\left(\frac{D-C}{1+e^{-2\left(\alpha +\beta \cdot x\right)}}\right)$$
where *S* is greyscale, *C* lower asymptote curve, *D* upper asymptote curve, *α* shift of the linear part of the sigmoid curve in the coordinate system, *β* slope of the linear part of the sigmoid curve, and *x* decimal logarithm of the analyte concentration. 

The limit of detection was defined as the concentration of an analyte corresponding to the maximum assay signal minus 3× standard deviation (SD) in accordance with the calibration curve (the blank was calculated from 3 parallel determinations with the absence of an analyte). The IC_50_ corresponded to the concentration of analyte giving 50% inhibition of the asymptotic maximum. The linear working range corresponded to the analyte concentration causing the 20–80% inhibition of the maximum assay signal.

### 3.7. Cross Interactions

Cross interactions (CR) were assessed visually after application of steroids at concentrations of 10 and 0.1 µg/mL by comparing the intensities of the TL bands of the individual steroids with the TL of the negative strip. The CL must always appear, if it does not appear, then the test is invalid and must be repeated. In the case of a competitive format, the TL appears only in the absence of analyte in the sample. If the analyte is present in the sample, then only the CL is visible. The steroids tested for CR are presented in Figure 1 and Appendix A. The strong CR was manifested by the disappearance of TL at both concentrations. A smaller interaction meant the disappearance of TL at a higher concentration and at a lower level only a weakening of TL. If the TL did not disappear or weaken, then no CR took place.

## 4. Results and Discussion

### 4.1. Immunochromatographic Test

After characterization of the RAb 212 antibody by indirect competitive ELISA (Huml et al. [7]), the ICT has been developed. The test strips have several parts (Figure 2A), i.e., membrane [39,40], conjugation, sample and absorbent pad. The membranes can be nitrocellulose, nylon, polyethylene or Teflon. The main criteria for their selection include the size of the pores that affect the course of the reaction and the flow rate of the reagents. The conjugation pad made of glass, polyester or cellulose fibers is part of a closed format, where a conjugate of NPs with antibodies is applied. The sample application pad is made of cellulose or glass fibers. Both of these pads also serve as a filter to retain unwanted larger particles from the sample. At the end of the strip, the absorbent pad accelerates the flow of reagents through the membrane and traps excess material. A disadvantage may be the need to change the buffers used to dilute immunoreagents [41]. Both indirect and direct competitive arrangement formats in open and closed format were tested. All experiments were performed in triplicate.

### 4.2. Indirect Format

Similarly to our previous report by Fojtíková et al. [8], coloidal gold particles and GAR antibody were used to prepare GAR-AuNPs conjugate. The following conditions were chosen for the open format: DAG concentrations in CL and RSA/ST-3 in TL, both 100 μg/mL, 2 µL RAb 212 applied to the strip, AE98 membrane, GFCP 103,000 glass fibers as sample pad and CFSP 223,000 absorbent pad. In the first experiment, the functionality of the RSA/ST-3 and GAR-AuNPs conjugate was tested. The GAR-AuNPs conjugate was applied in an amount of 3 and 5 µL. The best results were observed in an amount of 3 µL per strip (Appendix A). A calibration series with a visual detection limit of 0.5 ng/mL was subsequently set up for the conditions thus selected (Appendix A). Furthermore, CRs of the antibody with other steroids (structures in Figure 1 and Appendix A) were verified for this assay. 

In this experiment, steroids were applied at concentrations of 0.1 and 10 μg/mL. As can be seen from Figure 3A, 17α-methyltestosterone, methandienone, methyldihydrotestosterone, oxandrolone and oxymetholone showed strong CR. Boldenone, boldenone benzoate and testosterone showed lower CR. Dihydrotestosterone benzoate and testosterone decanoate did not cross-interact. CR was caused by a similar structure of steroids mainly in position C17. Therefore, the greatest reactivity was with steroids that have the same groups as ST (17α-methyl and β-hydroxyl) at C17 position. Conversely, those that have, for example, a longer aliphatic chain at this position did not respond. The RAb 212 antibody showed the same specificity during its characterization by indirect competitive ELISA (Huml et al. [7]). Thus, the specificity of the antibody was not affected by the format of the immunochemical method used. This conclusion was confirmed in previous reports [42,43].

After closing the system in boxes, the visual detection deteriorated the limit by one order of magnitude. It was, therefore, necessary to test the individual conditions that affect the speed and the quality of the result, which includes the intensity and sharpness of the lines and the sufficient buoyancy of the reaction mixtures. It was a matter of selecting a suitable membrane, individual components and the concentrations of immunoreagents. All tested materials are summed-up in Appendix A, and examples of some results are shown in Appendix A. Some materials have been retained for their suitability, others have been replaced by material that has resulted in a faster and higher-quality test result. The original and new materials are depicted in Appendix A. In all experiments, the concentration of DAG in CL was 100 μg/mL. Because the concentration of 100 µg/mL of RSA/ST-3 in TL taken from the open format was not sufficient, higher concentrations were tested, from which 300 µg/mL was selected for further experiments. For drying on a conjugation pad the following combinations of the amount of dried GAR-AuNPs and RAb 212 were tested (µL): 3:1, 3:2, 3:3, 4:1, 4:2, 4:3, 5:1, 5:2 and 5:3, from which a combination of 3 µL of GAR-AuNPs and 2 μL of RAb 212 was selected for further experiments. As can be seen in Appendix A, some membranes were unsuitable for this test (i.e., FF80HP, PRIMA 125) because the sample did not completely rise or it returned from the absorbent pad, and thus, the results could not be read despite the staining. Better, although not completely good, results were provided, e.g., by AE 100, where it was possible to read the results. However, the uptake was not sufficient and the reaction mixture returned from the absorbent pad, although this did not clearly affect the lines. Sufficient rise was achieved especially with AE 98 and FF120HP. The visual detection limits (VDLs) of these two membranes were then compared. Because this limit was 5 ng/mL for the AE 98 membranes used for the open format and 8 ng/mL for FF120HP, AE 98 was used in further experiments. Other significant differences were seen when testing the various absorbent pads (Appendix A). For these, there was a significant difference in thickness, which affected the rate of rise and the amount of reaction mixture captured. For some pads, absorption was insufficient and the mixture refluxed, leaving red streaks. Of all the pads tested, the Grade 320 and Grade 601 pads visually gave the best results. On the Grade 601 strips, the reaction mixture rose very slowly; even after 20 min, the result could not be read. Therefore, Grade 320 was chosen as a suitable pad for further tests, which was the strongest and, therefore, absorbed the most mixture; there was no backflow of immunoreagents and the result could be read already after 10 min. As a result, the strips were clean, free of reaction residue, and nothing affected the intensity of the lines. For other materials from Appendix A, the results differed mainly in the rate of rise in the mixture. 

Furthermore, various drying and reaction buffers were tested. Their composition is summarized in Appendix A. It was found that for the drying buffer, better results were obtained for buffers with the addition of trehalose, which increased the stability of the components of this buffer. For both buffers, the addition of Tween 20 detergent was necessary, without which the rate of rise was slower, the strips remained red colored, and thus, it was impossible to read the results. Tween 20 serves to reduce non-specific interactions, so it is an essential part of buffers. Furthermore, various combinations of pad drying procedures were tested—only the conjugation pad was dried in the buffer, only the sample pad, or both. The drying time was also examined—1 × 1 h, 2 × 1 h (this option was used in previous protocols described by us [8,26]) and 1 × 2 h. The best results were obtained with 0.2 mol/L borate buffer—0.1%, BSA—3%, trehalose—1% Tween 20 for a drying buffer and 0.1 mol/L borate buffer—1%, BSA—1%, PEG—1% Tween 20 for reaction buffer. Drying time was shortened from the original 2 × 1 h to 1 h with sufficient results. All original and newly selected immunoreagents and materials are summarized in Table 1. 

After selecting the appropriate test conditions, a calibration series with visual detection was set up limit around 1 ng/mL (Appendix A). For its more accurate determination, amounts of 0.5, 0.6, 0.7, 0.8, 0.9 and 1 ng/mL were further tested, of which 0.9 ng/mL was chosen as the VDL. Thus, the selection of suitable materials for the test reduced the VDL compared to 5 ng/mL, which were determined at the beginning of the experiments. 

The calibration series for this format was also evaluated using the computer program TotalLab. The resulting calibration curve of the logarithm of the concentration of ST on gray scale is displayed in Appendix A. Appendix A summarizes the characteristics of the standard ST curves (detection limit, IC_50_ value and linear working range). In the middle of the linear region is usually the concentration corresponding to the visual detection limit. In this case, this is equal to a concentration of 1 ng/mL, which corresponds to the detection limit determined visually (i.e., 0.9 ng/mL).

### 4.3. Direct Format

For the direct format, a conjugate of AuNPs with primary antibodies was newly synthesized (RAb-AuNPs). The functionality of this conjugate was tested in an open format, for which it was necessary to determine the amount of RAb-AuNPs conjugate, the concentration of GAR antibody in CL and RSA/ST-3 conjugate in TL. The tested amounts of RAb-AuNPs can be seen in Appendix A. An amount of 1.5 µL was selected as suitable for further tests. The concentration of GAR in CL was determined to be 150 µg/mL and the conjugate RSA/ST-3 in TL to be 200 µg/mL. Subsequently, a calibration series (Appendix A) was established, where it was found that the VDL is between concentrations of 0.5 and 1 ng/mL. Therefore, concentrations of 0.4, 0.5, 0.6, 0.7, 0.8, 0.9 and 1 ng/mL were tested, of which a concentration of 0.6 ng/mL was chosen as the VDL. The obtained ST standard curve was nearly identical to the standard curve obtained for indirect format, as can be seen from their analytical features shown in Appendix A.

Furthermore, CRs were verified for the same steroids at the same concentrations as for the indirect assay format (see Figure 3). From the strips, it was possible to read the same results as in the indirect format, i.e., 17α-methyltestosterone, methandienone, methyldihydrotestosterone, oxandrolone and oxymetholone showed strong CR. Boldenone, boldenone benzoate and testosterone showed lower CR; dihydrotestosterone benzoate and testosterone decanoate did not react.

After verifying the open format, the test was closed. Reagent concentrations in CL and TL were the same as for the open format; the other conditions were chosen to be the same as the indirect closed format. Only the amount of RAb-AuNPs to be dried on the conjugation pad was tested. Amounts of 1.5, 2, 2.5, 3, 3.5, 4, 4.5 and 5 μL were tested. As can be seen in Appendix A, CL are sufficiently intense for all options, but TL is sufficiently visible for up to 3.5 µL of RAb-AuNPs conjugate and was, therefore, used for further tests. This was followed by the determination of the VDL. From the calibration series (Appendix A), it was found that its value is below 1 ng/mL; therefore, concentrations of 0.5, 0.6, 0.7, 0.8, 0.9 and 1 ng/mL were tested of which 0.7 ng/mL was determined as the VDL. The achieved visual detection limit corresponds to the VDLs obtained in the other developed formats (visual detection limits range of 0.5—0.9 ng/mL) and is approximately an order of magnitude higher than the detection limit of ELISA (0.02 ng/mL) in our recent work by Huml et al. [7]. This difference corresponds to previously published studies [21,36,37]. None of the ICT test arrangements used resulted in a reduction in the detection limit compared to another format. The indirect closed ICT format will be more appropriate for routine analysis because its cost will be lower compared to the direct format. This format requires reduced amounts of specific (primary) antibody and a “universal label”—GAR-AuNPs—which is also suitable for other ICT methods that use a specific rabbit antibody as one of the immunoreactants.

The visual detectability of the developed methods is comparable to other ICTs for the analysis of low molecular weight compounds such as methiocarb, thiabendazole, forchlorfenuron, robenidine or imidocarb [21,36,44,45,46]. At the same time, the obtained detection limit is sufficient for the detection of commonly used concentrations of AASs for the illegal fortification of dietary supplements. To date, the ICT in the field of AAS detection has been published for methyltestosterone residues in animal feed with the cut-off value 2.5 ng/g [47], testosterone in milk samples with the cut-off value 0.5 ng/mL [48] or in drinking water with the cut-off value 5 ng/mL [49]. In any previously published work, it was not possible to test a group of 17α-methylated anabolics in one sample.

### 4.4. Spiked Food Supplements Testing

Food supplements are most often of powder or tablet nature, so it is necessary to extract them into a suitable solvent for testing. Ethanol was chosen as the solvent for sample extraction for the closed format of the indirect and direct assay settings, as well as for the ELISA [7]. For ICT, its effect on line intensity, especially on TL, was tested. EtOH was applied as 0.5, 1, 5, 10 and 20% solution diluted in reaction buffer for negative control. In the solutions, thus, prepared, the ST standard was subsequently diluted to a concentration of 1 ng/mL for positive control. In both cases, the respective TL of the strips was compared with either the TL on the strips with zero ST concentration (negative control) or with the TL on the strips with 1 ng/mL ST concentration (positive control) in the absence of EtOH. From the results, it was found that for the indirect format it is sufficient to dilute the EtOH to a 5% solution, but for the direct format, dilution is necessary to 1% solution (Appendix A). Ethanol extracts of food supplements (matrices—Section 3.2) were diluted 10×, 50×, 100× and 500×, i.e., to solutions with 10, 2, 1 and 0.5% EtOH content. When comparing TL intensities, it was found that for the indirect format, the lines are not affected by the 50× diluted (2% EtOH) matrix solution and for the direct format by the 100× diluted (1% EtOH) matrix solution (Appendix A). Therefore, the dilution of ethanol extracts of real samples to 1% solutions was chosen, where no effect of matrix or EtOH on any of the developed test formats should be recorded. 

Two sets of artificially contaminated samples were tested. Prior to the analysis, the effect of ten food supplement matrices (Appendix A) was tested [50]. No effect on the result was observed when samples 100× diluted were applied (Appendix A).

An assembled indirect and direct open format ICT assay was used for testing (Figure 4). In the first set, matrices of food supplements (Appendix A) were artificially contaminated (Section 3.2) with the same concentration of ST.

Ethanol extracts of the supplements (without ST) were diluted 100× (dilution recommended for sample preparation for assembled ICT assays) and analyzed by both assay formats. The resulting assay lines were visually compared to the assay line obtained for the negative control (experiment performed in reaction buffer only), and no difference in line intensity and quality was noted. Thus, the EtOH extracts of the supplements do not affect the assembled ICT assay formats, and the chosen sample preparation is suitable for further experiments. 

Subsequent analysis of 100× diluted EtOH extracts of artificially contaminated food supplements did not reveal a visually visible test line in any case, and all samples can be considered positive. Then, 250× dilute EtOH extracts were tested (at this dilution the concentration of ST in the EtOH extract is around the visual detection limit), and even in these analyses, the test line did not appear on all prepared strips, and thus, all samples can be marked as positive again (Appendix A).

The second set of artificially contaminated samples contained various combinations of AAS (Section 3.2) selected based on the article [51] and recommended combinations by the user [52,53] or AAS that did not cross-react at the concentrations we tested (Figure 5). On hundred-fold diluted EtOH extracts of the prepared samples were prepared, and the resulting test lines were visually observed. Mixtures 1–5 and the sample with ST did not show a test line, and samples can be considered positive. 

A visible test line was observed for samples containing boldenone and DHT. The resulting test lines were attenuated compared to the negative control test line. Thus, the antibody used shows CR at such high concentrations of AAS, which are used in real cases. This can be used if we use a two-strip box for sample analysis. In this box, one strip is used as a negative control and the other strip for sample analysis. It is, then, possible to visually monitor the attenuation of the test line in the sample compared to the test line in the control. In this case, the sample can be marked as positive. 

Contaminated dietary supplements found on the market have different contents of anabolic steroids [54,55,56]. However, the usual concentration is in the order of hundreds to tens of thousands of milligrams per kilogram. Therefore, this user-friendly method is very suitable for capturing suspicious samples.

## 5. Conclusions

Taken together, superior immunochromatographic tool for the determination of AAS containing 17α-methylated group was successfully developed. The presented study demonstrated the potential of using the colloidal gold nanoparticles as a tracer to provide visual evidence of the presence of AASs in dietary supplements within 10 min. To our knowledge, this is the first work using a single antibody in an immunochromatographic assay to determine many anabolic steroids for detection in dietary supplements. Semi-quantitative visual evaluation of the closed format of immunostrip provided the detection limit about 1 ng/mL. This concentration is far below the usually found concentration of AAS in contaminated dietary supplements. The functionality of the methods was verified by the analysis of real and artificially contaminated samples. Immunochromatographic assays are still solely screening methods. Therefore, it is always necessary to subject pre-positive test samples obtained by ICT to the other instrumental methods (GC-MS, HPLC-MS), which are, in fact, irreplaceable in the confirmation of the preliminary results, thus, obtained.

## Figures and Tables

**Figure 1 foods-10-00741-f001:**
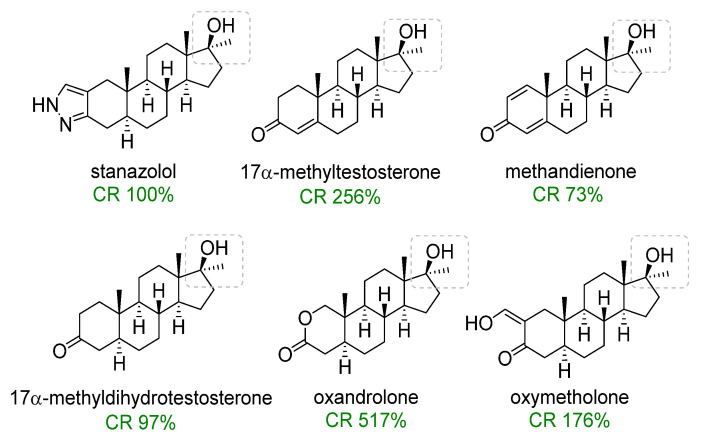
Molecular structures of major 17α-methylated anabolic steroids and their cross-reactivities (CR) (CR data from Huml et al. [7]).

**Figure 2 foods-10-00741-f002:**
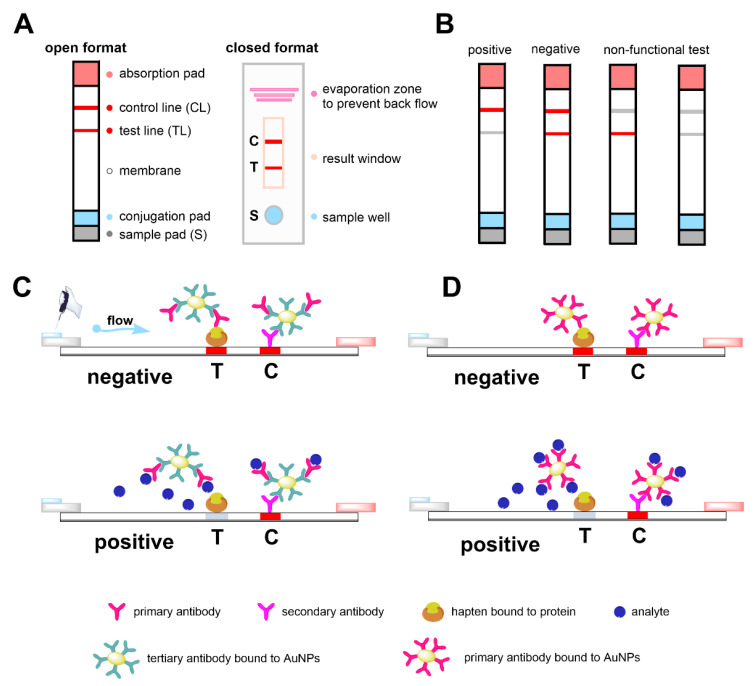
Test strips of immunochromatographic test (ICT) in competitive arrangement. Description of individual parts and representation of opened and closed format on panel (**A**); test results on panel (**B**); representation of indirect and direct ICT format on panels (**C**,**D**), respectively.

**Figure 3 foods-10-00741-f003:**
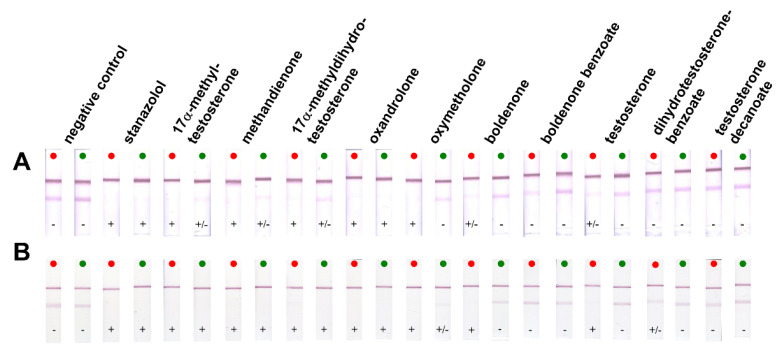
Cross-reactivity of selected steroids (Figure 1 and Appendix A) in indirect (panel (**A**)) and direct format (panel (**B**)) tested at 10 μg/mL (red dot) and 0.1 μg/mL (green dot). The tests were evaluated as negative (−), suspect (+/−) or positive (+).

**Figure 4 foods-10-00741-f004:**
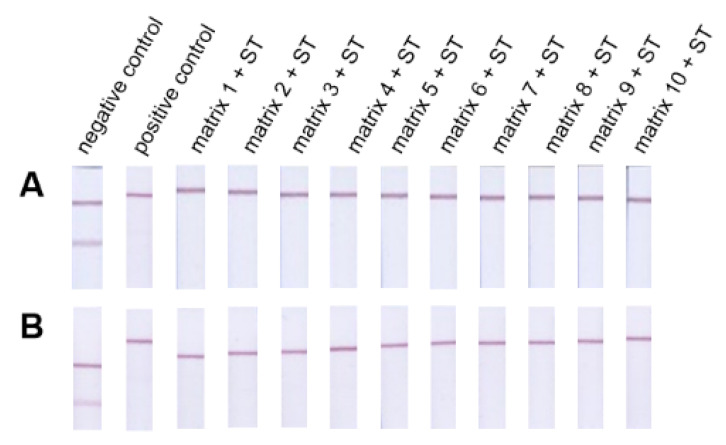
Spiked real supplements with ST (diluted 100×). The method was tested in both formats, i.e., indirect (panel (**A**)) and direct format (panel (**B**)).

**Figure 5 foods-10-00741-f005:**
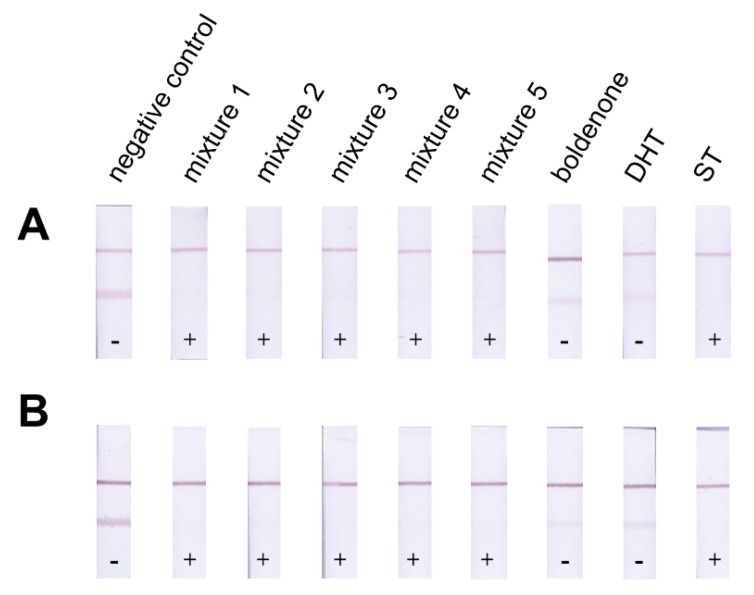
Result of artificially contaminated by indirect (panel (**A**)) and direct format (panel (**B**)).

**Table 1 foods-10-00741-t001:** A summary of optimized conditions for indirect ICT formats.

Parameter	Opened Format	Closed Format
Concentration of DAG in CL	100 μg/mL	100 μg/mL
Concentration of RSA/ST-3 in TL	100 μg/mL	300 μg/mL
Amount of GAR-AuNPs	5 μL	3 μL
Amount of RAb 212	2 μL	2 μL
Membrane *	AE 98	AE 98
Membrane pad *	HF000MC100	HF000MC100
Sample pad *	GFCP 103000	Grade 1281
Conjugation pad *	NA ^a^	Grade 6615
Absorption pad *	CFSP 223000	Grade 320
Drying buffer	NA ^a^	0.2 mol/L borate buffer—0.1%, BSA—3%, trehalose—1% Tween 20
Reaction buffer	0.1 mol/L borate buffer—1%, BSA—1%, PEG—1% Tween 20	0.1 mol/L borate buffer—1%, BSA—1%, PEG—1% Tween 20

* These conditions were used also for direct enclosed format. ^a^ Not applicable.

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
