# Peer review of "Tailor-Made Immunochromatographic Test for the Detection of Multiple 17α-Methylated Anabolics in Dietary Supplements"

_foods, 2021, doi:10.3390/foods10040741_

Round 1
Reviewer 1 Report
This manuscript doesn’t use statistical methods to analyze the data and support the conclusion, The author needs to provide more reliable data and analysis.
- Page 5 line 193, and page 6 line 200, should be “TL against CL”?
- Figure 3, 4, and 5 only show pictures of the strips, but how repeatable the results are? Strip test is not quantitative analysis, but it is still important to prove it is reliable based on statistic testing methods. These 3 figures are not enough to support the conclusion.
- Figure S7 doesn’t show any error bar for each data point, that is another big problem, the authors need to use statistics to show how reliable their method really is.
Author Response
The authors appreciate the time and effort that the reviewer has dedicated to providing valuable feedback on our manuscript. We are grateful to the reviewer for his comments on our paper. We have been able to incorporate changes to reflect the suggestions provided by the reviewer. Here is a point-by-point response to the reviewer's comments.
- Page 5 line 193, and page 6 line 200, should be “TL against CL”?
Response:
In the manuscript, the use of the term "TL versus TL" is correct. The authors modified text in the chapter 3.6. (Test evaluation) so that it is now easier for readers to understand.
- Figure 3, 4, and 5 only show pictures of the strips, but how repeatable the results are? Strip test is not quantitative analysis, but it is still important to prove it is reliable based on statistic testing methods. These 3 figures are not enough to support the conclusion.
Response:
The authors used a standard procedure to evaluate immunochromatographic strips - the intensity of the test lines was evaluated with the naked eye (Fan He et al., 2021; Youxue Wu et al., 2021; Lin Lu et al, 2021; Hendrickson et al., 2021). To verify repeatability, all experiments were performed in triplicate. In the manuscript (figures 3,4 and 5) only the results from one repetition are given for illustration. The authors added an explanation to the text of the manuscript (chapter 3.1).
Fan He, Jinyi Yang, Tingting Zou, Zhenlin Xu, Yuanxin Tian, Wenjia Sun, Hong Wang, Yuanming Sun, Hongtao Lei, Zijian Chen, Juewen Liu, Xuecai Tan, Yudong Shen, A gold nanoparticle-based immunochromatographic assay for simultaneous detection of multiplex sildenafil adulterants in health food by only one antibody, Analytica Chimica Acta, 1141, 2021, Pages 1-12, https://doi.org/10.1016/j.aca.2020.10.032.
Youxue Wu, Meijiao Wu, Cheng Liu, Yachen Tian, Shuiqin Fang, Hao Yang, Bin Li, Qing Liu, Colloidal gold immunochromatographic test strips for broad-spectrum detection of Salmonella, Food Control, 2021,108052,https://doi.org/10.1016/j.foodcont.2021.108052.
Lu Lin, Shanshan Song, Xiaoling Wu , Liqiang Liu , Hua Kuang, Jing Xiao and Chuanlai Xu, Determination of robenidine in shrimp and chicken samples using the indirect competitive enzyme-linked immunosorbent assay and immunochromatographic strip assay. Analyst, 2021, 146, 721-729 DOI: 10.1039/D0AN01783C.
Olga D. Hendrickson, Elena A. Zvereva, Natalia L. Vostrikova, Irina M. Chernukha, Boris B. Dzantiev, Anatoly V. Zherdev, Lateral flow immunoassay for sensitive detection of undeclared chicken meat in meat products, Food Chemistry, 2021,344,128598, https://doi.org/10.1016/j.foodchem.2020.128598.
- Figure S7 doesn’t show any error bar for each data point, that is another big problem, the authors need to use statistics to show how reliable their method really is.
Response:
On the reviewer's recommendation, the authors edited the Figure S7. Authors also added more statistical details for direct and indirect format (Table S2 with characteristics of stanazolol standard curves for both format). At the same time, they added details to chapter 2.7 (Test evaluation) including calculated values of the limit of detection, the IC50 and the linear working range.
Reviewer 2 Report
The manuscript “Tailor-made immunochromatographic test for the detection of multiple 17α-methylated AAS in dietary supplements” is generally well written and contains data of some relevance for a general readers as well as of high relevance for specialists in the topic. Although the subject of the paper could be of interest for the readers of the journal, the paper needs some corrections.
- Page 4, lines 113-125: I propose to list individual buffers (each in a separate line). The information presented in this way is not entirely clear.
- Page 4, lines 150-155: I propose to list individual mixtures (each in a separate line). The information presented in this way is not entirely clear.
- I suggest placing Figures and Tables closer to the discussed results.
- In my opinion, the discussion of the results is poor. I propose to refer to a few more literature items.
- Were the samples tested by other instrumental methods?
Author Response
The authors appreciate the time and effort that the reviewer has dedicated to providing valuable feedback on our manuscript. We are grateful to the reviewer for his comments on our paper. We have been able to incorporate changes to reflect most of the suggestions provided by the reviewer. Here is a point-by-point response to the reviewer's comments.
- Page 4, lines 113-125: I propose to list individual buffers (each in a separate line). The information presented in this way is not entirely clear.
- Page 4, lines 150-155: I propose to list individual mixtures (each in a separate line). The information presented in this way is not entirely clear.
Response:
The authors chose this method of presentation because it is commonly used in articles on the development of immunochemical methods and at the same time does not significantly extend the length of the text. This method is also used in articles already published in the journal Foods (Fu et al., 2021, Madrid et al., 2021).
Fu, X.; Chen, E.; Ma, B.; Xu, Y.; Hao, P.; Zhang, M.; Ye, Z.; Yu, X.; Li, C.; Ji, Q. Establishment of an Indirect Competitive Enzyme-Linked Immunosorbent Method for the Detection of Heavy Metal Cadmium in Food Packaging Materials. Foods 2021, 10, 413. https://doi.org/10.3390/foods10020413
Madrid, R.; García-García, A.; Cabrera, P.; González, I.; Martín, R.; García, T. Survey of Commercial Food Products for Detection of Walnut (Juglans regia) by Two ELISA Methods and Real Time PCR. Foods 2021, 10, 440. https://doi.org/10.3390/foods10020440
- I suggest placing Figures and Tables closer to the discussed results.
Response:
The template that must be used to submit an article to Foods journal no longer allows authors to move images and tables in the text. The text would then not be complete.
- In my opinion, the discussion of the results is poor. I propose to refer to a few more literature items.
Response:
The authors agree with the reviewer's recommendation and added further information and references to the chapter 3. Results and discussion.
- Were the samples tested by other instrumental methods?
Response:
The absence of anabolic steroids in tested samples (protein preparations) was verified with gas chromatography (Stepan et al. 2008) and certified anabolic androgenic steroid standards from Steraloids Inc. (Newport, USA) were used for contamination. Information was added to the chapter 2.3 Preparation of samples and matrices.
Stepan R, Cuhra P, Barsova S (2008) Comprehensive two-dimensional gas chromatography with time-of-flight mass spectrometric detection for the determination of anabolic steroids and related compounds in nutritional supplements. Food Addit Contam Part A Chem Anal Control Expo Risk Assess 25:557–565.
Reviewer 3 Report
Comments to the Authors:
The authors of this paper present a study on the Tailor-made immunochromatographic test for the detection of multiple 17α-methylated AAS in dietary supplements. Nevertheless, this work might be polished up taking into account the following suggestions
The topic of this paper is of scientific importance and within the scope of “foods”.
The abstract provides a clear overview of the experimental work and of the most relevant results obtained. The keywords are appropriate.
In the introduction section, the objectives of the experimental work are introduced. A critical presentation of the state-of-the-art was provided.
The “materials and methods” section accurately describes the statistical, analytical and experimental methods employed.
Tables and figures are clear and self-explanatory. The results are supported by literature contributions.
Therefore, the conclusions are supported by the results and the objectives of the work were fulfilled.
Overall, English language is acceptable.
Finally, I have only one observation in the title, please avoid the abbreviation of AAS.
Author Response
The authors appreciate the time and effort that the reviewer has dedicated to providing valuable feedback on our manuscript. We are grateful to the reviewer for his comments on our paper. Here is a response to the reviewer's comments.
- Finally, I have only one observation in the title; please avoid the abbreviation of AAS.
Response:
The authors agree with the reviewer's recommendation and changed the title of the article.
Round 2
Reviewer 1 Report
This time the data presentation and discussions are much better. I just have one additional question. The authors have used data obtained by either open format or closed format, but what is the criteria for choosing the format? Can authors add a brief summary regarding these two formats in the section of Conclusions?
Author Response
The authors thank the reviewer for commenting on our paper. As described in Chapter 1. Introduction (page 2, lines 67 - 74) of our manuscript, the open format was used for method development. The closed format is then suitable for the customer's own analysis of samples. And as stated in Chapter 5. Conclusion the developed closed format of immunostrip is suitable for the detection of AASs in contaminated dietary supplements. The authors added explanatory information to Chapter 5, line 460.